# Relationship among Gross Motor Function, Parenting Stress, Sense of Control, and Depression in Mothers of Children with Cerebral Palsy

**DOI:** 10.3390/ijerph18179285

**Published:** 2021-09-02

**Authors:** Eun-Young Park

**Affiliations:** Department of Secondary Special Education, Jeonju University, Jeonju 55069, Korea; eunyoung@jj.ac.kr; Tel.: +82-63-220-3186

**Keywords:** mothers of children with cerebral palsy, gross motor function, parenting stress, self-control, depression

## Abstract

The purpose of this study was to investigate the relationship among the gross motor function of children with cerebral palsy and parenting stress, sense of control, and depression in their mothers. Data were collected from 247 children with cerebral palsy and their mothers. To verify the relationship among variables, path analysis was performed. The control variables included the sex and age of the children. The proposed model showed good fit indices. Gross motor function had an indirect effect on parenting stress and depression and a direct effect on parenting stress and self-control (as parenting sense of control). Parenting stress had an indirect effect on depression and a direct effect on self-control and depression. This result suggests the importance of improving the gross motor function in children with cerebral palsy and self-control in the mothers, as well as decreasing parenting stress to reduce the level of the mothers’ depression. Considering the mediating effect of self-control on depression, programs designed to enhance self-control could be effective in decreasing depression in mothers of children with cerebral palsy.

## 1. Introduction

Parents of children with disabilities are known to experience more depression than parents of typically developing children in the process of accepting new and valuable information about their children’s disabilities [1]. Children with developmental disabilities, such as autism spectrum disorder, intellectual disabilities, and cerebral palsy (CP), display varying degrees of dysfunction in the acquisition of motor, cognitive, language, or social skills. CP is the most common marked motor developmental disorder. One study reported a range of depression incidence rates of 10% to 59% in mothers of children with autism spectrum, 10% to 79% in mothers of children with fragile X syndrome, and 30% to 38% in mothers of children with Down syndrome [2]. In children with cerebral palsy (CP), the long-term effects of the disability and accompanying problems are more pronounced [3]. CP refers to a group of disorders that are caused by non-progressive lesions or damage to the immature brain, leading to abnormal muscle tone and movement disorders. These disorders can be accompanied by sensory, cognitive, communication, and intellectual disabilities [4].

Health conditions in children with CP are complex, long-term, and often severe [5]. They negatively affect the daily life of children and their parents [6]. According to studies on mothers of children with CP, their degree of depression, anxiety, and stress is generally higher than that of mothers of typically developing children [7]. Ones et al. [8] reported that mothers of children with CP have a high incidence of depressive symptoms and a low quality of life. Manuel et al. [7] reported that 30% of mothers of children with CP have depressive symptoms. Depression requires special attention and support as it hinders the mother’s mental health and affects her parenting and treatment of children. 

Not all parents experience the same difficulties in caring for a child with CP [9]. Parental sense of control is an essential part of proper parental role performance. Parental sense of control refers to the expectation of a parent that he/she can raise children well and solve related problems [10]. In other words, parents with a high sense of control can cope with parenting stress by maximizing resources and maintaining this behavior [11]. Mothers with a high sense of control can appropriately use personal and social resources to establish a sensitive and harmonious relationship with infants with difficult temperaments. It was reported that no further efforts were made to interact with it [12]. 

Child-related variables, particularly the severity of the CP condition, have been considered as variables that affect stress. CP is the most common motor developmental disorder [13], and the atypical and abnormal movement patterns associated with the disorder make it difficult for parents to care for children with CP daily [14]. When children are at lower functional levels, their mothers report greater strain and lower quality of life [12,13,14,15,16]. Stress levels in mothers of children with CP were different according to the severity of motor dysfunction [15]. Caring for a child with limited self-mobility requires a high physical and psychological load, which leads to parenting challenges [16,17]. Although correlations have been found between children’s level of disability, parental depression, and parenting stress [18,19], their causal relationships remain unclear. The impact of the disability’s severity, perceived stress, and perceived social support on parental well-being has been examined, but the severity of disability is not a significant predictor of parental well-being [20]. 

Stress caused by children with disabilities harms the growth and development of their families, resulting in a decline in family function [4]. Therefore, there is a need to alleviate the depression of parents responsible for raising children with disabilities and enhance parental self-control for proper growth and healthy family life. Self-control is one of the health-related variables that can be impaired in mothers of children with CP. Previous research reported that self-control is an important personal resource to support the quality of life for parents of children with CP [6,21]. When caring for a family, self-control is the ability to care for oneself, respond to disruptive behavior and other care-related issues, and control negative thoughts and emotions arising from care [22]. To do so requires the individual to believe that they can cope with the situation and take breaks, delegate tasks, or ask for help when needed. This benefits both their well-being and how they care for others.

In this study, self-control was selected as an independent variable in the same context as self-efficacy in a mother’s role. Mastery is defined as a person’s belief that they can now control important situations that affect their lives [23]. Self-control is a measure of satisfaction and quality of life, and the sense of self-control perceived by an individual is a functional factor in maintaining a sense of well-being and efficacy under stressful or changing situations [24]. People with a high sense of self-control are less affected by stress as they successfully face challenges [25]. This study aims to determine the relationship among gross motor function, parenting stress, self-control, and depression in mothers of children with cerebral palsy.

## 2. Materials and Methods

### 2.1. Participants

This study was approved by the Research Ethics Board of Jeonju University (Jeonju University IRB-1041042-2013-1). The participants included 247 mothers of children with CP, of whom 138 had sons (55.9%) and 109 had daughters (44.1%). The children included in the study were undergoing rehabilitation treatment at one of 18 hospitals and community welfare centers located in various cities across South Korea. For the recruitment of participants, the investigators sent letters to colleagues, asking for referrals of eligible participants. The average age of the children was 8.51 years (SD = 4.32). Table 1 shows the children’s characteristics and their mothers’ education level and depression according to their characteristics. There was no significant difference in depression according to characteristics, with the type of CP being the only differentiating factor. Mothers of children with the ataxic type of CP showed a significantly lower level of depression than mothers of children with the dykentic type (*p* < 0.05). The sample size was confirmed to be appropriate for the study analysis based on the guideline for the ratio of the sample size to the number of free parameters. The sample size in this study was adequate based on the recommendation that 10–20 times as many cases as parameters are sufficient for significance testing of the model [26]. 

### 2.2. Measures

Control variables were controlled by including them in the path analysis model of this study as variables that affect endogenous variables. The child’s sex and age were the control variables in this study.

Mother depression was measured with CES-D [27]. The test developed by the Center for Epidemiologic Studies Depression (CES-D) [27] was used in this study. The CES-D was previously translated into Korean [28] and has been studied extensively. It has been shown to be a reliable and valid measure for depression in Korea. It consists of 20 items scored on a four-point Likert scale ranging from 0 (rarely or none of the time) to 3 (most or all the time). The examples of items were “I was bothered by things that usually bother me,” “I did not feel like eating,” and “I felt that I was as good as other people.” The Korean version of the CES-D has a scoring range of 0–60 points, where higher scores reflect more depressive symptoms. Cronbach’s α coefficient was 0.909 and ranged from 0.892 to 0.925.

The children’s gross motor function was measured with GMFCS. The gross motor function classification system (GMFCS) is a tool developed to assess motor impairment in children with CP. It was used to evaluate the grand motion function of children with CP. The GMFCS classifications are as follows: Level 1 can walk without any restrictions; Level 2 can walk with restrictions; Level 3 can walk with a cane, crutches, or walker without trunk support; Level 4 has restrictions but can move using an electric wheelchair or other means of mobility device; and Level 5 is a case where mobility is severely limited even with the use of assistive devices [29]. The inter-rater reliability was reported as 0.84 [30].

The mothers’ sense of control was measured with the Mastery Scale [31]. This scale was also translated into Korean [32]. Five items were reversed, and each one is scored with a range of 7–28 points, where higher scores indicate stronger feelings of control. The validity and reliability of the Mastery Scale were confirmed [33]. The examples of the items were: “No way I can solve some of the problems I have,” “Sometimes I feel that I am being pushed around in life,” and “I have little control over the things that happen to me.” Cronbach’s α coefficient was 0.805 and ranged from 0.765 to 0.840.

The Korean Parenting Stress Index (K-PSI) was used to assess the parenting stress of mothers of children with CP. K-PSI is the standardized Korean version of the PSI developed in the United States by Lee and Chung [34]. The examples of the items were: “My child rarely does things for me that make me feel good,” “My child smiles at me much less than I expected,” and “I feel trapped by my responsibilities as a parent.” In the standardization study, the internal consistency coefficient was 0.56–0.95, and the test–retest reliability was 0.50–0.90 [34]. Item analysis was performed in the previous study on parents of children with CP [35]. In the present study, Cronbach’s α coefficient was 0.910 and ranged from 0.891 to 0.926.

Data regarding GMFCS were collected by physical therapists who had treated children with CP for more than 6 months. Mothers of children with CP provided information about the Mastery Scale, K-PSI, and depression. 

### 2.3. Statistical Analysis

Descriptive statistics were used to find the general characteristics of the study’s participants. An independent t-test and one-way analysis of variance were performed to find the differences in depression according to the related variables. Spearman’s correlation test was employed to test the multicollinearity among variables, an outcome that was not observed in this study. The statistical significance level was set as α = 0.05.

Structural equation modeling (SEM) was employed in this study to examine the relationships among the CP children’s gross motor function and the mothers’ self-control, parenting stress, and depression. The mediating effect of self-control and parenting stress on depression was also examined. 

Root mean square error of approximation (RMSEA) values < 0.05, 0.06–0.08, 0.08–0.10, and > 0.1 indicate good, reasonable, mediocre, and poor fit, respectively [36]. Normed fit index (NFI) and comparative fit index (CFI) > 0.90 also indicate a good fit [36,37]. 

## 3. Results

### 3.1. Multicollinearity Test

The inter-correlations of gross motor function, parenting stress, self-control, and depression were analyzed using Pearson’s correlation coefficients, provided in Table 2. The correlations were in the expected directions. Multicollinearity was not detected, as the bivariate correlations did not exceed 0.80 [38]. 

### 3.2. Preliminary Path Analysis

Preliminary path analyses were performed to determine significant variables for model identification. The direct effects of gross motor function on depression were not significant (β = −0.013, *p* = 0.808). The proposed path model of this study is shown in Figure 1. 

### 3.3. Model Fit 

Absolute-fit indices and incremental-fit measures were used to categorize the identified fit indices to evaluate the model fit. Absolute-fit indices measure how well a model fits the data without comparison to a baseline model, and incremental-fit measures determine how well a model fits compared to a baseline model. *χ*^2^ and RMSEA among absolute-fit indices and NFI and CFI among incremental-fit measures were used to verify the model’s adequacy in this study. Because *χ*^2^ is affected by sample size, it was only reported in this study. The proposed model showed excellent fit indices (Table 3). Generally, a good model fit is indicated by RMSEA scores lower than 0.10 and NFI and CFI scores greater than 0.9.

### 3.4. Path Coefficients

The standardized regression weight among variables is shown in Table 4. Regression coefficients were statistically significant (*p* < 0.05) except between variables and control variables such as the children’s age and sex. The regression weight between parenting stress and self-control was largest among significant coefficients (β = −0.509). The regression weight between gross motor function and self-control was smallest among significant coefficients (β = 0.185). 

### 3.5. Direct and Indirect Effects of Variables

Table 5 presents the total, direct, and indirect effects of the variables. The independent variable of depression was 35.5%, explained by children’s sex, children’s age, gross motor function, parenting stress, and self-control. Parenting stress and self-control had significant direct and indirect effects on depression. The independent variable of self-control was explained by predictor variables of children’s sex, children’s age, gross motor function, and parenting stress, and the amount of explained variance was 33.2%. The gross motor function had both direct and indirect effects on self-control. Of the variance of parenting stress, 4.3% was explained by the predictor variables of children’s age, children’s sex, and gross motor function. 

## 4. Discussion

The purpose of this study was to investigate the relationship between the gross motor function of children with CP and the self-control, parenting stress, and depression in their mothers through path analysis. For this purpose, data were gathered to measure the gross motor function of children with CP and the self-control, parenting stress, and depression of their mothers; the data were then analyzed. Depression and parenting stress in the mothers of children with CP was reported to be high, and a high sense of self-control was required to lower it. The relationship between the child’s gross motor function and the parenting stress, self-control, and depression of the mother can provide basic data for developing psychological and educational programs for mothers of children with CP by notifying the influence direction and moderating influence between variables. The implications of this study based on major findings are as follows. 

First, the relationship between the functional level of children with CP and parenting stress was consistent with that in previous studies. The severity of CP is a major factor influencing parenting stress [12,13,14,15,16]. Among the child-related variables, the severity of CP has been considered a variable that affects stress. Mothers of children with CP at lower functional levels report greater stress and lower quality of life than those at higher functional levels [12,13,14,15,16]. Parenting challenges arise when providing care for children with CP. In this study, the relationship between the gross motor function level of children with CP and parenting stress was investigated through path analysis. As a result, when the gross motor function level of children with CP increased by 1 standard deviation, the mothers’ parenting stress decreased by 0.202. This is consistent with the results of previous studies. The gross motor function of children with CP affects self-control. When the gross motor function level of children with CP increased by 1 standard deviation, the mothers’ self-control increased by 0.185.

Second, parenting stress was associated with parents’ depression. Parenting stress has been reported as a major factor influencing the development of children in the context of families [39]. The parents of children with CP have been reported to have higher parenting stress than parents of typically developing children [40]. Parents of children with CP often perceive themselves to be their children’s advocates who must ensure that they receive optimal services [41], and more than 60% of them are vulnerable to parenting stress [18]. Information on the factors affecting stress may be meaningful to help provide appropriate services. This information may be helpful to professionals who can then provide adjusted services based on the needs of the family [42]. The results of this study showed that alleviating parenting stress in mothers of children with CP was associated with reducing depression.

Third, self-control has a direct relationship with the mother’s depression and can moderate the relationship of variables on depression. Self-control is related to well-being, psychological and physical health, task performance, and emotions. Those with high self-control had fewer psychological difficulties such as depression, anxiety, and stress [43,44]. Self-control also has a positive relationship with interpersonal relationships, and people with higher self-control have better interpersonal skills and higher satisfaction with interpersonal relationships than those with low levels [45,46]. Self-controlled behavior builds strength, confidence, comfort, and independence in terms of the ability to direct one’s life. Studies have shown a correlation between self-control and behavioral problems in a wide range of problem areas, such as cancer, anxiety, nightmares, medical procedures, exam anxiety, depression, sleep problems, enuresis, encopresis, and stuttering [47]. Based on the results of this study, when the self-control level increased by 1 standard deviation, the mothers’ depression decreased by 0.314. When parenting stress level increased by 1 standard deviation, the mothers’ depression increased by 0.363. The results of this study showed that the gross motor function of children with CP has a significant direct and indirect effect on self-control. The relationship of gross motor function of children with CP on self-control was mediated by parenting stress, and the relationship of the gross motor function of children with CP on depression was mediated by self-control. 

Self-control is defined as the ability to regulate and control inner impulses, wishes, emotions, and behaviors [48]. It is a set of skills that allows individuals to form behaviors using their free will while replacing one type of behavior with another that is more desirable [49,50]. Through self-control, it is possible to understand and control inner needs and emotions and to achieve positive results by adjusting to the environment [51]. Self-control is derived from the assumption that human behavior is goal-oriented and is always undergoing a process of change and development. It is especially important when a person needs to learn new patterns of behavior or make decisions or when previous behaviors are no longer effective [49]. It is therefore a flexible, self-regulating ability that changes undesirable thoughts, emotions, and behaviors to adapt to the environment more effectively and predicts success through long-term goal achievement [52]. The results of this study showed that self-control could mediate the relationship between gross motor function and parenting stress on depression. This result is like that of Stewart et al. [53], which indicated that a sense of control acts as a modulating variable in the relationship between downward social comparison and subjective well-being in the elderly. The results further suggested that therapeutic interventions to increase the functional level of children with CP, as well as support programs to reduce mothers’ parenting stress and increase the level of self-control of mothers, are needed to reduce the depression level of mothers of children with CP.

The results also confirmed the relationship among the gross motor function of children with CP, parenting stress, self-control, and depression of their mother. More specifically, self-control mediated the relationship of gross motor function and parenting stress on depression. Mothers of children with CP with high self-control were found to have low levels of parenting stress and depression. This lends credence to the belief that self-control can act as a protective variable against subjective well-being and depression, even if an individual is placed in difficult situations. Recent systematic review and meta-analysis reported that interventions to improve psychological well-being in the parents of children with CP was effective [54]. 

The limitations of this study were as follows. First, although children diagnosed with CP have some common characteristics, differences according to the type of CP do exist. This study could not identify the differences according to the type of CP. Although the sample size is adequate, the number of children with the dykentic type of CP was much smaller, and there are likely to be other confounders that contribute to these results. Considering that the main participants of this study were the spastic type, it is highly likely that the characteristics of children with spastic CP were reflected. Therefore, in future studies, investigating the relationship between the variables according to the type of CP will be necessary. Second, this study is a cross-sectional correlation study, and there are limitations in revealing the long-term relationship between variables. Therefore, in future studies, it is necessary to investigate the long-term relationship between variables through intervention and to enable a comprehensive interpretation through longitudinal studies.

## 5. Conclusions

The hypothesis of this study was confirmed based on our study and analysis of children with CP. Mothers with increased self-control were observed to have less depression when considering the relationship between stress and severity of CP. The results of this study also suggested that it is necessary to develop policies and services that will contribute to the improvement of psychosocial adjustment. These could include rehabilitation intervention to maintain or improve children’s functioning level or an empowerment program to increase the self-control of mothers of children with CP. 

## Figures and Tables

**Figure 1 ijerph-18-09285-f001:**
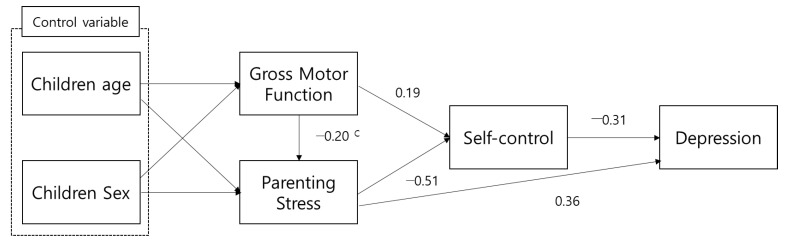
Proposed path model. ^c^ = standardized coefficient.

**Table 1 ijerph-18-09285-t001:** Participants’ characteristics (children with CP and their mothers).

Characteristics	Frequency	%		Depression		
			M	SD	*t/F*	*p*
*Children with CP*						
Sex						
Male	138	55.9	35.98	8.58	−0.391	0.696
Female	109	44.1	36.43	9.59		
Type of CP						
Spastic	209	84.6	36.11	9.13 ^ab^	3.158	0.044
Dykentic	30	12.1	38.40	8.23 ^a^		
Ataxic	8	3.2	29.50	5.63 ^b^		
GMFCS level						
Level 1	47	19.0	33.83	8.18	1.501	0.190
Level 2	36	14.5	36.11	10.44		
Level 3	26	10.5	36.92	7.64		
Level 4	36	14.5	34.78	7.71		
Level 5	102	41.5	37.66	9.49		
*Mothers*						
Education level						
Graduate school	5	2.0	32.20	7.85	0.502	0.681
University	160	64.8	36.68	9.59		
High school	45	8.2	35.67	8.76		
Middle school	2	0.8	38.50	14.85		
Missing data	35	14.2				

Note: GMFCS = gross motor function classification system. Mean of depression sharing the same letter subscript (^a,b^) within characteristics are not significantly different at the *p* < 0.05 level.

**Table 2 ijerph-18-09285-t002:** Correlation between variables.

Category	Gross Motor Function	Parenting Stress	Self-Control	Depression
Gross motor function	-	−0.203 **	0.289 **	−0.176 **
Parenting stress		-	−0.547 **	0.535 **
Self-control			-	−0.513 **

Note: ** *p* < 0.01.

**Table 3 ijerph-18-09285-t003:** Fit index of the proposed model.

*df*	*χ* ^2^	NFI ^a^	CFI ^b^	RMSEA ^c^
6	11.063	0.953	0.977	0.059

Note. ^a^ Normed fit index; ^b^ comparative fit index; ^c^ root mean square error of approximation.

**Table 4 ijerph-18-09285-t004:** Path coefficients of the conceptual model.

Path	β ^a^	Standard Error	Critical Ratio	*p*
Children age → Gross motor function	−0.084	0.005	−1.335	0.182
Children sex → Gross motor function	−0.101	0.041	−1.592	0.111
Children age → Parenting stress	−0.018	0.244	−0.28	0.78
Children sex → Parenting stress	0.030	2.123	0.477	0.633
Gross motor function → Parenting stress	−0.202	3.284	−3.205	0.001
Parenting stress → Self-control	−0.509	0.013	−9.559	<0.001
Gross motor function → Self-control	0.185	0.692	3.484	<0.001
Self-control → Depression	−0.314	0.131	−5.142	<0.001
Parenting stress → Depression	0.363	0.033	5.938	<0.001

^a^ Standard coefficients.

**Table 5 ijerph-18-09285-t005:** Total, direct, and indirect effects of variables.

Predictor Variables	Independent Variables	Total Effect	Direct Effect	Indirect Effect	*R^2^*
Children’s sex	Gross motor function	−0.101	−0.101	-	0.017
Children’s age		−0.084	−0.084	-	
Children’s sex	Parenting stress	0.050	0.030	0.020	0.043 *
Children’s age		0.000	−0.018	0.017	
Gross motor function		−0.202 *	−0.202 *	-	
Children’s sex	Self-control	−0.044	-	−0.044	0.332 *
Children’s age		−0.015	-	−0.015	
Gross motor function		0.288 *	0.185 *	0.103 *	
Parenting stress		−0.509 *	−0.509 *	-	
Children’s sex	Depression	0.032	-	0.032	0.355 *
Children’s age		0.005	-	0.005	
Gross motor function		−0.164 *	-	−0.164 *	
Parenting stress		0.523 *	0.363 *	0.160 *	
Self-control		−0.314 *	−0.314 *	-	

* *p* < 0.05.

## Data Availability

The data presented in this study are available on request from the corresponding author.

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
