# Peer review of "Relationship among Gross Motor Function, Parenting Stress, Sense of Control, and Depression in Mothers of Children with Cerebral Palsy"

_ijerph, 2021, doi:10.3390/ijerph18179285_

Round 1

Reviewer 1 Report

Interesting topic to explore.  Parents of children diagnosed with cerebral palsy are in need of additional support and the research is of great value.  Thank you. 

I am interpreting the method as data was collected once from all participants.  This can indicate a relationship (either direct or indirect between motor function of child and parent stress, self control and depression and allow for different relationships to be observed through modelling but the term 'effect' suggests that each child is measured more than once with variables adjusted and other confounding effects controlled or adjusted for.  I realise that this is not possible in this research design but would suggest you consider using terms 'relationship', 'impact' or 'contribution' rather than 'effect'. Children with cerebral palsy often have other complex health conditions as the author notes in the introduction  and hence other variables would be 'in play' in these results. 

Table 1: Dykentic instead of dyskinetic, check spelling throughout. 

Table 1: It would be helpful to have common symptoms for each type included in the introduction and dyskinetic interpreted in the Discussion for  the reader (the number is much smaller and although the sample size is adequate there is likely other confounders that contribute to these results).

Discussion: Again 'effect' and 'causal' should be replaced with words as noted earlier as the tests were not repeated and variables were not controlled to make this assumption.

Line 200: Table is numbered 4 instead of 5.

Line 216: Causal? Also check use of effect throughout.

Conclusion: 'increases in self control' - I am not sure this is reflective of the study as the self control was not actually increased during the study.  Perhaps word as something along the lines of 'parents with increased self-control were observed to have less depression when taking into account stress and severity of CP'? 

Overall, valuable piece of research that can be applied to interventions to assist parents of children with CP. Thank you for the read.

Author Response

Dear reviewer

Thank you very much for giving me the opportunity to revise our manuscript. I am grateful to you for insightful suggestions and comments. I have carefully reviewed the comments and incorporated them to strengthen our manuscript.

In the revised manuscript, I have highlighted in blue where we have made changes; please, note that I removed track changes that show all my edits.

The details of the changes are provided below this letter.

Thank you again for your time and efforts to review my manuscript and to provide insightful suggestions and edits.

Comment #1: I am interpreting the method as data was collected once from all participants. This can indicate a relationship (either direct or indirect between motor function of child and parent stress, self-control and depression and allow for different relationships to be observed through modelling but the term 'effect' suggests that each child is measured more than once with variables adjusted and other confounding effects controlled or adjusted for. I realize that this is not possible in this research design but would suggest you consider using terms 'relationship', 'impact' or 'contribution' rather than 'effect'. Children with cerebral palsy often have other complex health conditions as the author notes in the introduction and hence other variables would be 'in play' in these results. 

Response #1: Thank you for your comment. The term of effect has been revised into relationship or impact. Because the terms of direct and indirect effect are statistical words, they are remained. 

Throughout manuscript

Comment #2: Table 1: Dykentic instead of dyskinetic, check spelling throughout. 

Response #2: Thank you’re your correction. There were corrected throughout the manuscript.

Comment #3: Table 1: It would be helpful to have common symptoms for each type included in the introduction and dyskinetic interpreted in the Discussion for the reader (the number is much smaller and although the sample size is adequate there is likely other confounders that contribute to these results).

Response #3: Your comment has been reflected in the manuscript.

CP is the most common motor developmental disorder [13] and the atypical and abnormal movement patterns associated with the disorder make it difficult for parents to care for children with CP daily [14].

Although the sample size is adequate, the number of children with the dykentic type of CP was much smaller, and there are likely to be other confounders that contribute to these results. Considering that the main participants of this study was spastic type, it is highly likely that the characteristics of children with spastic CP were reflected. Therefore, in future studies, investigating the relationship between the variables according to the type of CP will be necessary.

Comment #4: Discussion: Again 'effect' and 'causal' should be replaced with words as noted earlier as the tests were not repeated and variables were not controlled to make this assumption.

Response #4: The term of effect or causal in discussion has been revised into relationship or impact.

Throughout manuscript

Comment #5: Line 200: Table is numbered 4 instead of 5.

Response #5: It has been corrected.

Comment #6: Line 216: Causal? Also check use of effect throughout.

Response #6: It has been revised.

The results of this study showed that self-control could mediate the relationship of gross motor function and parenting stress on depression.

Comment #7: Conclusion: 'increases in self-control' - I am not sure this is reflective of the study as the self-control was not actually increased during the study.  Perhaps word as something along the lines of 'parents with increased self-control were observed to have less depression when taking into account stress and severity of CP'? 

Response #7: The sentence has been revised according to comment.

Mothers with increased self-control were observed to have less depression when considering the relationship between stress and severity of CP.

Comment #8: Overall, valuable piece of research that can be applied to interventions to assist parents of children with CP. Thank you for the read.

Response #8: Thank you for your valuable comments.

Reviewer 2 Report

Thank you for the opportunity to review this manuscript. The author conducted a path analysis to examine the causal relationships between the gross motor function of children with CP and the parenting stress, self-control, and depression of their mothers. The sample included 247 children with CP and their mothers. It was reported that gross motor function had an indirect effect on stress and depression, and a direct effect on stress and self-control. Stress had an indirect effect on depression and direct effect on depression and self-control. Unfortunately, as it stands now, this paper contains too many issues that prevent me from recommending publication. Issues noted include:

Lines 30-31: The claim that the problems for children with CP compared to those of other children with disabilities is not justified.

Lines 33-34: Do you mean “movement disorders”?

Line 40: Check formatting of in-text citations. Do you mean “Ones and Yilmaz”?

Lines 58-61: This seems somewhat redundant given the description of CP in the first paragraph.

Lines 65-66: Expand on the description of this large scale study.

The author does not provide a strong rationale for conducting this study. While it is mentioned that the “causal relationships remain unclear” among these variable, a path analysis is not really a remedy to this issue. Path analyses just clarify correlations and can be used to indicate the strength of causal hypotheses.

Line 85-90: The reasoning for this decision to substitute SC for SE is not strong. The work cited (Hamama et al., 2008) for the claim that SE and SC are interconnected does not actually make this claim. The work cited (Hwang, 2020) for the claim that SC is a function of SE under certain situations does not make this claim. Further strengthening of the rationale for the inclusion of SC as a substitute for SE seems to be needed.

Lines 176-178: Citation for interpretation of fit indices? For RMSEA, values between .08 and .10 are typically “acceptable” but not “good.” For NFI and CFI, .95 or higher would be considered good fit.

Line 209-211: Explain how this data allows us to “[develop] psychological and educational programs for mothers of children with CP.”

Provide a discussion of the implications for these findings.

Provide a discussion of the study’s limitations.

In the conclusion, the author discusses the need for the development of policies and services that support psychosocial adjustment. There is a large body of literature on supports for psychosocial adjustment. A review of this literature and evidence-based recommendations for what can be done to support psychosocial adjustment for parents of children with CP would enhance the discussion/conclusion section greatly.

There are many mechanical writing errors throughout the manuscript that impact its overall quality/readability.

Author Response

Dear reviewer

Thank you very much for giving me the opportunity to revise our manuscript. I am grateful to you for insightful suggestions and comments. I have carefully reviewed the comments and incorporated them to strengthen our manuscript.

In the revised manuscript, I have highlighted in blue where we have made changes; please, note that I removed track changes that show all my edits.

The details of the changes are provided below this letter.

Thank you again for your time and efforts to review my manuscript and to provide insightful suggestions and edits.

Comment #1: Lines 30-31: The claim that the problems for children with CP compared to those of other children with disabilities is not justified.

Response #1: Description has been inserted.

Comment #2: Lines 33-34: Do you mean “movement disorders”?

Response #2: It has been corrected into movement disorders.

Children with developmental disabilities, such as autism spectrum disorder, intellectual disabilities, and cerebral palsy (CP), display varying degrees of dysfunction in the acquisition of motor, cognitive, language, or social skills. CP is the most common marked motor developmental disorder. One study reported a range of depression incidence rates of 10% to 59% in mothers of children with autism spectrum, 10% to 79% in mothers of children with fragile X syndrome, and 30% to 38% in mothers of children with Down syndrome [2]. In children with cerebral palsy (CP), the long-term effects of the disability and accompanying problems are more pronounced. CP refers to a group of disorders that are caused by non-progressive lesions or damage to the immature brain, leading to abnormal muscle tone and movement disorders. These disorders can be accompanied by sensory, cognitive, communication, and intellectual disabilities [3].

Comment #3: Line 40: Check formatting of in-text citations. Do you mean “Ones and Yilmaz”?

Response #3: It was corrected into Ones et al. (8)

Ones et al. [8] reported that mothers of children with CP had a high incidence of depressive symptoms and a low quality of life.

Comment #4: Lines 58-61: This seems somewhat redundant given the description of CP in the first paragraph.

Response #4: The sentence has been modified to reduce redundant.

Comment #5: Lines 65-66: Expand on the description of this large scale study.

Response #5: Description has been expanded.

Child-related variables, particularly the severity of the CP condition, have been considered as variables that affect stress. CP is the most common motor developmental disorder [13] and the atypical and abnormal movement patterns associated with the disorder make it difficult for parents to care for children with CP daily [14]. When children are at lower functional levels, their mothers report greater strain and lower quality of life [12-16]. Stress levels in mothers of children with CP were different according to the severity of motor dysfunction [15]. Caring for a child with limited self-mobility requires a high physical and psychological load, which leads to parenting challenges [16-17]. Although correlations have been found between children’s level of disability, parental depression, and parenting stress [18,19], their causal relationships remain unclear. The impact of the disability’s severity, perceived stress, and perceived social support on parental well-being have been examined, but severity of disability was not a significant predictor of parental well-being [20].

Comment #6: The author does not provide a strong rationale for conducting this study. While it is mentioned that the “causal relationships remain unclear” among these variable, a path analysis is not really a remedy to this issue. Path analyses just clarify correlations and can be used to indicate the strength of causal hypotheses.

Response #6: I agreed with your comment and the term of effect or causal have been revised into relationship.

Comment #7: Line 85-90: The reasoning for this decision to substitute SC for SE is not strong. The work cited (Hamama et al., 2008) for the claim that SE and SC are interconnected does not actually make this claim. The work cited (Hwang, 2020) for the claim that SC is a function of SE under certain situations does not make this claim. Further strengthening of the rationale for the inclusion of SC as a substitute for SE seems to be needed.

Response #7: The background has been expanded.

In this study, self-control was selected as an independent variable in the same context as self-efficacy. This is an effective substitute, as the concept of self-efficacy and self-control were perceived to be interconnected in the 1990s [25]. Bandura [26] believed that one of the most effective ways to develop a strong sense of efficacy was through the mastery of one’s own experiences. Mastery is defined as a person’s belief that they can now control important situations that affect their lives [27]. Self-control is a measure of satisfaction and quality of life, and the sense of self-control perceived by an individual is a functional factor in maintaining a sense of well-being and efficacy under stressful or changing situations [28]. People with a high sense of self-control are less affected by stress as they successfully face challenges [29]. This study aims to determine the relationship among gross motor function, parenting stress, self-control, and depression in mothers of children with cerebral palsy.

Comment #8: Lines 176-178: Citation for interpretation of fit indices? For RMSEA, values between .08 and .10 are typically “acceptable” but not “good.” For NFI and CFI, .95 or higher would be considered good fit.

Response #8: Citation has been inserted (R. P. Sarmento and V. Costa, “Confirmatory factor analysis--a case study,” 2019, https://arxiv.org/abs/1905.05598).

Root mean square error of approximation (RMSEA) values < 0.05, 0.06–0.08, 0.08–0.10, and >0.1 indicate good, reasonable, mediocre, and poor fit, respectively [40]. Normed fit index (NFI) and comparative fit index (CFI) > 0.90 also indicate a good fit [40, 41].

Comment #9: Line 209-211: Explain how this data allows us to “[develop] psychological and educational programs for mothers of children with CP.”

Response #9: Explain has been inserted.

The relationship among the child’s gross motor function, and the parenting stress, self-control, and depression of the mother can provide basic data for developing psychological and educational programs for mothers of children with CP by notifying the influence direction and moderating influence between variables.

Comment #10: Provide a discussion of the implications for these findings.

Response #10: Implications have been inserted.

Implications of this study based on major findings are as follows.

First, the relationship between the functional level of children with CP and parenting stress was consisted with previous studies. The severity of CP is a major factor influencing parenting stress [12-16].

Second, parenting stress was associated with parents’ depression.

Third, self-control has a direct relationship with the mother’s depression and could moderate the relationship of variables on depression.

Comment #11: Provide a discussion of the study’s limitations.

Response #11: Limitations have been inserted.

The limitations of this study were as follows. First, although children diagnosed with CP have some common characteristics, differences according to the type of CP do exist. This study could not identify the differences according to the type of CP. Although the sample size is adequate, the number of children with the dykentic type of CP was much smaller, and there are likely to be other confounders that contribute to these results. Considering that the main participants of this study was spastic type, it is highly likely that the characteristics of children with spastic CP were reflected. Therefore, in future studies, investigating the relationship between the variables according to the type of CP will be necessary. Second, this study is a cross-sectional correlation study, and there are limitations in revealing the causal relationship between variables. Therefore, in future studies, it is necessary to investigate the causal relationship between variables through intervention and to enable a comprehensive interpretation through longitudinal studies.

Comment #12: In the conclusion, the author discusses the need for the development of policies and services that support psychosocial adjustment. There is a large body of literature on supports for psychosocial adjustment. A review of this literature and evidence-based recommendations for what can be done to support psychosocial adjustment for parents of children with CP would enhance the discussion/conclusion section greatly.

Response #12: Conclusion has been revised according to comment.

The hypothesis of this study was confirmed based on our study and analysis of children with CP. Mothers with increased self-control were observed to have less depression when considering the relationship between stress and severity of CP. The results of this study also suggested that it is necessary to develop policies and services that will contribute to the improvement of psychosocial adjustment. These could include rehabilitation intervention to maintain or improve children’s functioning level or an empowerment program to increase the self-control of mothers of children with CP.

Comment #13: There are many mechanical writing errors throughout the manuscript that impact its overall quality/readability.

Response #13: English proof reading has been performed to correct mechanical writing errors.

Reviewer 3 Report

The manuscript presented is of enormous interest and relevance and it is worth paying attention to the impact on mothers of caring for children with cerebral palsy.

The following are some aspects that could improve the work presented:
- In the introduction the authors dwell on the aspects of stress, self-control and depression but it is extremely surprising that they do not devote a single word to motor function. In this sense, it would be relevant for the authors to contextualize this aspect and the influence it may have on the psychological aspects or the daily life of the parents.
- Similarly, the authors need to clarify the objective that appears in the last paragraph of the introduction. It does not correspond to the title or to what they later explore.
- In the methodology, the authors should explain how the sample was recruited, the procedure for administering the tools, and the ethical aspects.
- In the discussion, the authors should include the limitations of the study carried out and the future lines of research based on the work done.
- The authors should rework the conclusions to adapt them to the object of the research carried out. 

Author Response

Dear reviewer

Thank you very much for giving me the opportunity to revise our manuscript. I am grateful to you for insightful suggestions and comments. I have carefully reviewed the comments and incorporated them to strengthen our manuscript.

In the revised manuscript, I have highlighted in blue where we have made changes; please, note that I removed track changes that show all my edits.

The details of the changes are provided below this letter.

Thank you again for your time and efforts to review my manuscript and to provide insightful suggestions and edits.

Comment #1: The manuscript presented is of enormous interest and relevance and it is worth paying attention to the impact on mothers of caring for children with cerebral palsy.

Response #1: Thank you for your valuable comments.

Comment #2: In the introduction the authors dwell on the aspects of stress, self-control and depression but it is extremely surprising that they do not devote a single word to motor function. In this sense, it would be relevant for the authors to contextualize this aspect and the influence it may have on the psychological aspects or the daily life of the parents.

Response #2: According to your comment, description related to motor function has been inserted in introduction.

CP is the most common motor developmental disorder [13] and the atypical and abnormal movement patterns associated with the disorder make it difficult for parents to care for children with CP daily [14]. When children are at lower functional levels, their mothers report greater strain and lower quality of life [12-16]. Stress levels in mothers of children with CP were different according to the severity of motor dysfunction [15]. Caring for a child with limited self-mobility requires a high physical and psychological load, which leads to parenting challenges [16-17]. Although correlations have been found between children’s level of disability, parental depression, and parenting stress [18,19], their causal relationships remain unclear. The impact of the disability’s severity, perceived stress, and perceived social support on parental well-being have been examined, but severity of disability was not a significant predictor of parental well-being [20].

Comment #3: Similarly, the authors need to clarify the objective that appears in the last paragraph of the introduction. It does not correspond to the title or to what they later explore.

Response #3: Objective has been revised.

This study aims to determine the relationship among gross motor function, parenting stress, self-control, and depression in mothers of children with cerebral palsy.

Comment #4: In the methodology, the authors should explain how the sample was recruited, the procedure for administering the tools, and the ethical aspects.

Response #4: Methodology section has been revised according to your comments.

This study was approved by the Research Ethics Board of Jeonju University (Jeonju University IRB-1041042-2013-1). The participants included 247 mothers of children with CP, of which 138 had sons (55.9%) and 109 had daughters (44.1%). The children included in the study were undergoing rehabilitation treatment at one of 18 hospitals and community welfare centers located in various cities across South Korea. For recruitment of participants, the investigators sent letters to colleagues, asking for referrals of eligible participants.

Data regarding GMFCS were collected by physical therapists who had treated children with CP for more than 6 months. Mothers of children with CP provided information about the Mastery Scale, K-PSI, and depression.

Comment #4: In the discussion, the authors should include the limitations of the study carried out and the future lines of research based on the work done.

Response #5: Limitations has been inserted and recommendations for future research has been inserted.

The limitations of this study were as follows. First, although children diagnosed with CP have some common characteristics, differences according to the type of CP do exist. This study could not identify the differences according to the type of CP. Although the sample size is adequate, the number of children with the dykentic type of CP was much smaller, and there are likely to be other confounders that contribute to these results. Considering that the main participants of this study was spastic type, it is highly likely that the characteristics of children with spastic CP were reflected. Therefore, in future studies, investigating the relationship between the variables according to the type of CP will be necessary. Second, this study is a cross-sectional correlation study, and there are limitations in revealing the causal relationship between variables. Therefore, in future studies, it is necessary to investigate the causal relationship between variables through intervention and to enable a comprehensive interpretation through longitudinal studies.

Comment #6: The authors should rework the conclusions to adapt them to the object of the research carried out. 

Response #6: Conclusion has been revised.

The hypothesis of this study was confirmed based on our study and analysis of children with CP. Mothers with increased self-control were observed to have less depression when considering the relationship between stress and severity of CP. The results of this study also suggested that it is necessary to develop policies and services that will contribute to the improvement of psychosocial adjustment. These could include rehabilitation intervention to maintain or improve children’s functioning level or an empowerment program to increase the self-control of mothers of children with CP.

Round 2

Reviewer 2 Report

Thank you for the opportunity to review this revised paper. I am delighted to see the improvements made to the manuscript, which I applaud the authors for. In particular, I appreciate the additions to the methods section as well as the discussion of limitations. However, there continues to be a number of issues that prevent me from recommending publication at this time without significant edits.

Lines 33-34: As argued, individuals with CP displaying varying degrees of dysfunction. There is no citation to qualify the claim that is being made here that the long-term problems are more pronounced when compared to these other disorders that can significantly impair an individual’s functioning.

Again, check your formatting of in-text citations. When there are two authors, you name both authors. “Et al” is used for three or more authors.

As noted in my previous review, two articles (Hamama et al., 2008; Hwang, 2020) were cited to argue something that they did not in fact claim. The authors did not address these inaccuracies.

As previously noted, further strengthening of the rationale for the inclusion of SC as a substitute for SE seems to be needed. The revised version included a sentence about Bandura believing that developing SE occurred through the mastery of SC. While this provides minimal theoretical justification for the substitution, it doesn’t appear sufficient for making a convincing case for this decision. Are there studies that help to justify this?

The explanation provided for how this data allows us to “[develop] psychological and educational programs for mothers of children with CP” is insufficient. The discussion briefly mentions that “basic data” can be used “by notifying the influence direction and moderating influence between variables.” It’s not clear to me, based on this limited description, how the data can be used for program development.

As previously noted, there is a large body of literature on supports for psychosocial adjustment. A review of this literature and evidence-based recommendations for what can be done to support psychosocial adjustment for parents of children with CP would enhance the discussion/conclusion.

There continues to be many mechanical writing errors throughout the manuscript that impact its overall quality/readability.

Author Response

Dear reviwer

Thank you very much for giving me the opportunity to revise our manuscript. I am grateful to you for insightful suggestions and comments. I have carefully reviewed the comments and incorporated them to strengthen our manuscript.
In the revised manuscript, I have highlighted in blue where we have made changes; please, note that I removed track changes that show all my edits.
The details of the changes are provided below this letter.
Thank you again for your time and efforts to review my manuscript and to provide insightful suggestions and edits.

Comment #1: Lines 33-34: As argued, individuals with CP displaying varying degrees of dysfunction. There is no citation to qualify the claim that is being made here that the long-term problems are more pronounced when compared to these other disorders that can significantly impair an individual’s functioning.

Response #1: I have inserted the reference. 

Comment #2: Again, check your formatting of in-text citations. When there are two authors, you name both authors. “Et al” is used for three or more authors.

Response #2: I have checked it. 

Comment #3: 

As noted in my previous review, two articles (Hamama et al., 2008; Hwang, 2020) were cited to argue something that they did not in fact claim. The authors did not address these inaccuracies.

As previously noted, further strengthening of the rationale for the inclusion of SC as a substitute for SE seems to be needed. The revised version included a sentence about Bandura believing that developing SE occurred through the mastery of SC. While this provides minimal theoretical justification for the substitution, it doesn’t appear sufficient for making a convincing case for this decision. Are there studies that help to justify this?

Response #3: According to editor and reviewer comments, I have revised my manuscript to delete the self-efficacy. I used the term of self-control. Self-control does not mean as a kind of self-efficacy any more. 

Comment #4: 

The explanation provided for how this data allows us to “[develop] psychological and educational programs for mothers of children with CP” is insufficient. The discussion briefly mentions that “basic data” can be used “by notifying the influence direction and moderating influence between variables.” It’s not clear to me, based on this limited description, how the data can be used for program development.

As previously noted, there is a large body of literature on supports for psychosocial adjustment. A review of this literature and evidence-based recommendations for what can be done to support psychosocial adjustment for parents of children with CP would enhance the discussion/conclusion.

Response #4: I review the literature and reflected in manuscript discussion section. 
